# Citation Classics in Consumer Neuroscience, Neuromarketing and Neuroaesthetics: Identification and Conceptual Analysis

**DOI:** 10.3390/brainsci11050548

**Published:** 2021-04-27

**Authors:** Pablo Sánchez-Núñez, Manuel J. Cobo, Gustavo Vaccaro, José Ignacio Peláez, Enrique Herrera-Viedma

**Affiliations:** 1Joint-PhD Programme in Communication, Department of Audiovisual Communication and Advertising, Faculty of Communication Sciences, Universidad de Málaga, 29071 Málaga, Spain; 2Center for Applied Social Research (CISA), Universidad de Málaga, 29071 Málaga, Spain; fabianvaccaro@uma.es (G.V.); jipelaez@uma.es (J.I.P.); 3Instituto de Investigación Biomédica de Málaga (IBIMA), 29010 Málaga, Spain; 4Department of Computer Science and Engineering, School of Engineering, Universidad de Cádiz, 11202 Cádiz, Spain; manueljesus.cobo@uca.es; 5Department of Languages and Computer Science, Higher Technical School of Computer Engineering, Universidad de Málaga, 29071 Málaga, Spain; 6Andalusian Research Institute on Data Science and Computational Intelligence, Department of Computer Science and AI, University of Granada, 18071 Granada, Spain; viedma@decsai.ugr.es

**Keywords:** Bibliometrix, consumer behaviour, consumer psychology, consumer neuroscience, H-index, highly cited papers (HCPs), scientometrics, science mapping analysis, SciMAT

## Abstract

Neuromarketing, consumer neuroscience and neuroaesthetics are a broad research area of neuroscience with an extensive background in scientific publications. Thus, the present study aims to identify the highly cited papers (HCPs) in this research field, to deliver a summary of the academic work produced during the last decade in this area, and to show patterns, features, and trends that define the past, present, and future of this specific area of knowledge. The HCPs show a perspective of those documents that, historically, have attracted great interest from a research community and that could be considered as the basis of the research field. In this study, we retrieved 907 documents and analyzed, through H-Classics methodology, 50 HCPs identified in the Web of Science (WoS) during the period 2010–2019. The H-Classic approach offers an objective method to identify core knowledge in neuroscience disciplines such as neuromarketing, consumer neuroscience, and neuroaesthetics. To accomplish this study, we used Bibliometrix R Package and SciMAT software. This analysis provides results that give us a useful insight into the development of this field of research, revealing those scientific actors who have made the greatest contribution to its development: authors, institutions, sources, countries as well as documents and references.

## 1. Introduction

Consumer neuroscience, neuromarketing, and neuroaesthetics are three subfields of neuroscience, and play a key role in the design’s products, experiences, or services since they are concerned with how our brain perceives, process and reacts to the various stimuli presented in our environment [1,2,3,4].

Consumer neuroscience is the discipline that investigates the neural correlates of consumer decisions with a clear focus on progress in basic scientific understanding of the brain [5], whereas neuromarketing is the discipline that tries to use the methodologies and insights from consumer neuroscience in applied research and business applications [6], and neuroaesthetics, which the main goal is to characterize the neurobiological foundations and evolutionary history of the cognitive and affective processes involved in aesthetic experiences and artistic and creative activities as well as to recognize which are the subjective ratings and universal standard and patterns of beauty [7,8], by using all the power of functional magnetic resonances (fMRI), electroencephalography (EEG), pupillometry or eye-tracking techniques, among others to analyze what happens in our brain during the contemplation of beauty and ugliness [9].

Previous research has demonstrated how consumer neuroscience, neuromarketing and neuroaesthetics are fundamental when it comes to determining new commercial viewpoints, analyzing the consumer in a non-intrusive way without asking questions or market research, refining the user experience, reinforcing the brand image, or crafting a marketing message that attracts quality leads and improves conversion rates, among others [10,11]. Furthermore, from the entrepreneur’s point of view, consumer neuroscience, neuromarketing and neuroaesthetics improve risk reduction, since experiences, products, and services are designed and developed according to the tastes of individuals. The objectives of these neuroscientific techniques are, precisely, to know how the nervous system translates the high number of stimuli to people exposed in different contexts, as well as to understand the consumer’s attitudes and motivations so that helps to anticipate and select the advertising and marketing strategy that causes the greatest emotional impact on individuals [12,13,14]. Some of the recent studies in these subfields of neuroscience have not been without controversy regarding the relationship between consumer neuroscience, neuromarketing and neuroaesthetics. More specifically, in the field of neuroaesthetics, some different currents or detractors advocate a separation between empirical aesthetics and neuroaesthetics or demand a formal definition of what is one and what is the other [15]. However, their view is not necessarily widely shared by the research community, and that, though there may be good arguments to merge consumer neuroscience, neuromarketing and neuroaesthetics, the fields currently are separated.

Within this context, synthesizing the results of recent and relevant research becomes a critical duty to advance in a specific research line. Through H-Classics methodology [16], we can summarize the scientific production in a knowledge domain like consumer neuroscience, neuromarketing and neuroaesthetics. The H-Classics methodology [16] is based on the study of “citation classic”, also called “literacy classic” or “classic article” [17]. The H-Classics is an extension of the H-Index metric that is often used to quantify an individual’s research output [18]. Some authors have established a criteria threshold in the number of articles designated, limiting the list to the 50 [19] or 100 [20] most cited ones, or restricting the selection to those articles that have been cited at least 400 times [21]. Nevertheless, Martinez et al. [22] proposed that the selection of classic articles should be based on two parameters:-The computing of the H-Index [18]. Burrell points out that the H-Index identifies the most productive core of an author’s output in terms of the most cited papers [23].-The computing of the H-Core, also called Hirsch Core. For this core, consisting of the first h papers, Rousseau [24] introduced the term H-core, which can be considered as a group of high-performance publications concerning the scientist’s career [23,25].

The study of classic articles is crucial because they attract the interest of the scientific community and are therefore considered the “gold bars of science” [26,27] and permits analysis of the past, present, and future of a specific area of knowledge. The advantages of H-Classics are that, through a structured process, we can review the research work with the highest impact in a systematic, transparent, and reproducible way [16,28,29,30]. H-Classics method comprises the collection in a single process of the number of papers available in each field and their impact. It is also simple to compute and sensitive to alterations among areas in the impact of papers [16]. Due to such virtues, several studies focused on highly cited papers (HCPs) and citation classics have been published in different disciplines. Highly cited papers are papers that perform in the top 1% based on the number of citations received when compared to other papers published in the same field in the same year. Some outstanding examples are H-Classics studies in rheumatology [30], biology [31], intelligent transportation systems [32], social work [33], implant dentistry, periodontics, and oral surgery [29], aggregation operators in group decision making [34], or fuzzy decision making [28], and the rest.

To the best of our knowledge, there is no study covering the recent research about consumer neuroscience, neuromarketing and neuroaesthetics. The HCP of the consumer neuroscience, neuromarketing and neuroaesthetics research field has not been analyzed and explored yet from a Citation Classics scientometric perspective.

According to the aim stated above, some aspects can be analyzed by identifying the set of HCP: (I) the HCP distribution during the period studied; (II) the most productive journals, authors, institutions and countries; and, (III) the main topics covered by the papers detected.

Due to this high social importance, the present study aims at identifying the Highly Cited Papers (HCP) in consumer neuroscience, neuromarketing and neuroaesthetics disciplines, to deliver a summary of the academic work during the last decade in this area and to show tendencies, practices, methods, and findings that could be the basis for future advances in the discipline.

This study aims to answer the following research questions:➢RQ1. What are the leaders and knowledge hubs (most relevant authors, affiliations, countries, and sources) in consumer neuroscience, neuromarketing and neuroaesthetics?➢RQ2. What are the disruptive documents and sources (most relevant cited papers, references, and sources) in consumer neuroscience, neuromarketing and neuroaesthetics?➢RQ3. What is the conceptual structure (motor themes, and emerging or declining themes) in scientific publications about consumer neuroscience, neuromarketing and neuroaesthetics?

To do so, the paper is set out as follows: Section 2, defines the materials, methods, and the approach used in the analysis, Section 3 presents the results of our analysis, and, finally, Section 4 concludes with the discussion, conclusion, and upcoming investigation approaches.

## 2. Materials and Methods

### 2.1. Bibliographic Database and Data Acquisition

The H-Classics analysis was realized based on academic publications related to consumer neuroscience, neuromarketing, and neuroaesthetics. The source of information was the Web of Science (WoS) database.

The WoS database (Clarivate Analytics, Philadelphia, PA, USA), was founded by Eugene Garfield, one of the precursors of informetrics. The WoS is a collection of databases of bibliographic references and citations from periodicals that collect information from 1900 to the present. The choice of the WoS as the data source was made based on two main characteristics of the database: it provides numerous analysis tools for processing the data and it offers highly accurate and reliable research information [30,33].

In this research, we gathered 907 scientific documents ranging from 2010 to 2019.

907 results were checked for applicability in the present work. Authors considered the period 2010–2019 the one that covers the most recent academic production in this research area by, establishing the last decade as a timespan, to analyze the most recent and innovative innovations and discoveries in consumer neuroscience, neuromarketing and neuroaesthetics. The specific search equation was formulated according to the search logic of the WoS database. The terms selected for the search equation were keywords related to consumer behaviour and neuroscience (neuromarketing, neuroaesthetics, and consumer neuroscience). Choosing all document types and scientific work written in any language. Table 1 illustrates the query design. The retrieved WoS dataset and the citation report are available at the Zenodo repository [35].

### 2.2. H-Classics Methodology

According to Martinez et al. [16], there are four key phases to follow for carrying out the identification procedure of HCPs of a research area applying the H-Classics concept:(1)Bibliographic database and source of information selection to retrieve the study sample. Some commonly used examples are the PubMed database, Google Scholar database, Scopus database, or WoS database. For the reasons explained above, the database chosen in this study is the WoS.(2)Delimit the WoS research area. Due to the interdisciplinary nature of the study subject, this discipline focuses on many sub-areas/categories in WoS (such as COMMUNICATION, HEALTH SCIENCES, MARKETING, BUSINESS, MANAGEMENT, NEUROSCIENCE, NEUROIMAGING, SOCIAL SCIENCES, etc.). Due to this, the subject category has not been delimited in WoS, as it could produce a skewed and incomplete result. To properly delimit the research area, in this study we have elaborated the query that is presented in Table 1.(3)Computing the H-Index. The computation of the H-Index of the document retrieval is completed by establishing a ranking of the papers according to their citations in the research area. The WoS database provides us with filtering tools (Citation Report) to easily compute the H-Index of the study topic.(4)Computing the H-Core. The H-Core and consequently the HCPs in this field of research were retrieved through the automatic function provided by Web of Science, called “Create Citation Report”. The tool provides the H-Index of the HCPs, known as H-Core.

Following the example:

If we have retrieved N articles and their respective citations subject to scientific category of A, we could also calculate the H-Index of category A as we calculate the H-index of a researcher [16] i.e.,


*A paper P of scientific category A is considered an H-Classic of A if and only if P is inside of the H-core of A.*


In such a way,


*H-Classics of a research area A could be defined as the H-core of A that is composed of the H highly cited papers with more than H citations received.*


### 2.3. Citation Report and Record Count

Table 2 shows the Citation Report and the Record Count resulting from the WoS querying. The total publications gathered (907), combined a sum of 9931 times cited and produced an average of 10.95 citations per paper. The H-Index is the same as 50, which means that 50 studies have received at least 50 citations. Consequently, the H-Classics selection (50 H-Index documents) is the study sample of this work.

### 2.4. Scientometric and Science Mapping Analysis Tools

To perform the H-Classics Analysis, we have used the following scientometric and science mapping analysis tools:Bibliometrix version 3.0.2, designed by Aria and Cuccurullo [36], is an open-source tool, developed in R. It is designed for informetrics research including all the main bibliometric methods of analysis (co-citation, coupling, and scientific collaboration analysis). The package allows importing bibliographic data from the main scientific databases such as Scopus, WoS, PubMed, or Cochrane, among others. Bibliometrix has been frequently used by researchers in various fields of knowledge such as educational technology research [37], Big Data and Social Media [19], or infectious diseases [20], along with others. We used Bibliometrix to perform the annual production of H-Classics, the percentage of citations of papers as well as to detect the most relevant authors, documents, affiliations, and countries.SciMAT version 1.1.04, developed by Cobo, López-Herrera, and Herrera-Viedma [21,22], is an open-source science mapping software tool that incorporates methods, algorithms, and measures for all the steps in science mapping workflow, from preprocessing to the visualization of the results. SciMAT has been recently used by several researchers in fields of knowledge such as library and information science [38], scientometrics and COVID-19 (or coronavirus or SARS-CoV-2) research [39] or artificial intelligence and machine learning [40]. We used SciMAT to study the incoming and outgoing keywords, the conceptual structure of science and key themes over time as well as the identification of developing or decreasing topics.

## 3. Results

### 3.1. Main Information about the Collection

Table 3 presents the summarized main results of the H-Classics analysis. The table displays key information about the bibliographic data retrieved and several sub-tables, such as annual scientific production, top manuscripts per number of citations, most productive authors, most productive countries, total citation per country, most relevant sources (journals), and most relevant keywords.

Additionally, various diverse co-authorship indices are exposed. In specific, the Authors per Article index is calculated as the ratio between the total number of authors and the total number of articles. The Co-Authors per Articles index is measured as the average number of co-authors per article. In this circumstance, the index reflects the author’s appearances while for the “authors per article” an author, even if he published more than one article, is counted only on one occasion (Authors per Article index ≤ Co-authors per Article index). The CI or Collaboration Index (Total Authors of Multi-Authored Articles/Total Multi-Authored Articles) is a Co-authors per Article index calculated only using the multi-authored article set [41].

### 3.2. Distribution of Publications by Year and Record Count

Figure 1 shows the distribution of publications during the period 2010–2017. During the first five-year period of the study (2010–2014) there was a scientific production in H-Classics of 40 papers (*n* = 40, 80%): seven papers published in 2010, 12 papers in 2011, 11 papers in 2012, seven papers in 2013 and three papers in 2014. During the second five-year period (2015–2019) there was a scientific production in H-Classics of 10 papers (*n* = 20, 20%): six papers published in 2015, one paper published in 2016, three papers published in 2017, 0 papers published in 2018 and 0 papers published in 2019. As it can be seen, the first five years (2010–2014) were the most productive in H-Classics, with the highest scientific production, and 2011 the year with the highest sum of publications (12). The Figure shows results up to 2017, in this sense, papers published in consumer neuroscience, neuromarketing and neuroaesthetics during 2018 and 2019 have not reached the highly cited category (HCPs) by other authors, for this reason, they are not represented.

### 3.3. Average Citations per Year

Table 4 show the Average Citations per Year. The results show that 2010, with 7 H-Classics papers, was the year with the highest average percentage of citations per paper (115.14), considering that had 10 years of citation ahead of it (which is the length of the study period). However, the most satisfactory year in terms of total citations per year (TCpY) was 2017, with the highest peak (represented in Figure 2), and with 3 H-Classics papers published and an average of 38.11 total citations per year (TCpY) and only 3 citable years (CY) (the remainder of the study period until 2019).

### 3.4. Most Local Cited Sources (from Reference Lists), Most Relevant Sources, Source Local Impact, and Source Dynamics

A source is a journal/book/conference proceeding series/etc. which published one or more documents included in our bibliographic collection. In our collection, we have a total of 36 different sources.

Table 5 shows Most Local Cited Sources (from Reference Lists). Local citations measure how many times a document included in this collection have been cited by the documents also included in the collection. A cited source is a journal/book/conference proceeding series/etc. included in at least one of the reference lists (bibliography) of the document set. In our collection, we have 1428 cited sources included in the 50 document bibliographies. In this case, NEUROIMAGE stands out in 1st position with 182 articles, in 2nd position JOURNAL OF NEUROSCIENCE with 117 articles, in 3rd position we found NEURON with 111 articles, in 4th position we found JOURNAL OF CONSUMER RESEARCH with 99 articles and in 5th position we found SCIENCE with 87 referenced articles.

Table 6 shows the most relevant sources, as well as the source local impact, the initial year of publication of the manuscripts in the sources, the total number of citations (TC), the number of publications (NP), the number of articles, and various indexes and metrics of scientific productivity such as the H-Index [18], the G-Index [42] and the M-Index [43]. G-Index is a variant of the H-Index that, in its calculation, gives credit for the most highly cited papers in a data set and the G-Index is always the same as or higher than the H-Index and M-Index is another variant of the H-Index that displays H-Index per year since first publication [44].

The most relevant source was in 1st place *Journal of Consumer Psychology* (with six published articles, a total of 739 citations and starting to publish articles from this selection in the year 2010), followed in 2nd position FRONTIERS IN HUMAN NEUROSCIENCE (with four articles, a total of 304 citations and starting to publish articles from this selection in the year 2011), in 3rd position JOURNAL OF MARKETING RESEARCH (with four published articles, a total of 389 citations and beginning to publish articles from this selection in the year 2015), in 4th place NEUROIMAGE (two articles, a total of 238 citations and beginning to publish articles from this selection in the year 2011) and in 5th place PSYCHOLOGY & MARKETING (with two published articles, a total of 259 citations and beginning to publish articles from this selection in the year 2011).

The distribution frequency of articles (Figure 3) indicates the sources dealing with the issue and related topics and it calculates yearly published documents of the top sources (Table 7). Between 2011 and 2014, it was substantial growth in the number of publications on the subject. The graph shows the result of the Loess regression. As variables, it includes the number and the publication time of the source under study. This method allowed the function to assume a limitless distribution, that is, it permits the function to adopt values below zero if the data is near to zero. It contributed to a better graphic result and highlights the break in the history of the publications [45].

### 3.5. Most Relevant Authors, Author’s Impact, Most Local Cited Authors, Top-Authors’ Production over the Time and Corresponding Author’s Country

Table 8 shows the most relevant authors and their local impact. It quantifies an individual author’s contribution to a published set of papers. The two authors who had the major scientific production in H-Classics were in 1st position, the researcher A. Chatterjee with four papers in H-Classics and a total of 438 citations and 2nd position the researcher M. Nadal, also with four papers in H-Classics and a total of 272 citations.

Publication Year Start indicates the first year the author published in (in the set of included studies). Chatterjee and Nadal started publishing papers that are part of the H-Classics collection in 2011.

Table 9 shows the generated frequency table of the Most Local Cited Authors. Local Cited Authors measure how many times an author included in this collection have been cited by the authors also included in the collection. Chatterjee, besides being one of the authors with more scientific production and with the highest impact was also an author of reference for the authors of H-Classics appearing in other works (being cited more than 54 times).

Figure 4 and Table 10 show the Author’s Production Over Time. The function visually represented calculates and plots the authors’ production (in terms of the number of publications, and total citations per year) over time.

Total citations represent the number of times each manuscript has been cited and Total citations per year represent the yearly average number of times each manuscript has been cited.

The three authors who had a longer and more consistent trajectory in H-Classics were: Chatterjee (2011–2016), Nadal (2011–2016), and Dimoka (2010–2105). Chatterjee was the author, who besides having one of the longest and most consistent trajectories in H-Classics during the period studied, had H-Classics publications almost every year of those five years (four publications). Venkatraman was the author, who, despite having one of the shortest trajectories in H-Classics during the studied period, presented several works in 2015 that make him the author with the highest TC/TCpY (two H-Classics publications freq., 229 citations and total citations per year of 38.167).

Figure 5 shows the data visualization of the countries of origin of the most relevant corresponding authors of the H-Classics selection. The correspondence author (APC) or designated co-author is the person who will serve as a representative on behalf of all co-authors, by establishing contact during the process of submission, review, and final editing of the manuscript with the editor-in-chief and associate editors of a particular journal. MCP represents the inter-country and SCP represents the intra-country collaboration indices corresponding author of each manuscript. MCP indicates, for each country, the number of documents in which there is at least one co-author from a different country. In contrast, the SCP represents scientific papers that have been published and all co-authors of the manuscript have an affiliation from the same country. As can be seen in Figure 5 and Table 11, the USA stands out from the rest of the selection with a total of 19 corresponding authors in 19 different articles and a frequency of 38 participations, followed by Germany in 2nd place with a total of five corresponding authors in five articles and a frequency of 10 corresponding authors, and in 3rd place the United Kingdom with a total of five corresponding authors in five articles and a frequency of 10 corresponding authors. The USA had a rather high publication rate in intra-collaboration, more than half of their scientific output in H-Classics. Also, to a lesser extent Germany and United Kingdom. However, Canada, France, Spain, and Australia had a presence in production in terms of corresponding authors, but we observe how the work they have produced has been through the authors’ inter-collaboration.

### 3.6. Most Relevant Affiliations (Organizations)

Table 12 shows the most relevant affiliations, the frequency distribution of affiliations of all co-authors for each paper. Two international reputation and excellence rankings of worldwide universities were used to compare the results: ARWU World University Rankings 2019 [46] and QS World University Rankings 2019 [47].

Temple University was the American university with the largest volume of articles (12), followed by the American universities University of Pennsylvania (2nd position and 10 articles), Duke University (3rd position and eight articles), Emory University (6th position and five articles), The University oF California, Los Angeles (7th position and five articles) and Columbia University (9th position and four articles). The rest of the institutions were European organizations belonging to Germany (Freie Universität Berlin in 4th position and eight articles), Denmark (Copenhagen Business School in 5th position and five articles), the UK (the University of Oxford in 8th position and five articles) and the Netherlands (Erasmus Universiteit Rotterdam in 10th position and 4 articles).

All the research centres and organizations selected in the table had a presence in at least one of the two prestigious international university rankings (QS/ARWU) and all of them occupied relevant positions within the university reputation rankings.

### 3.7. Country Scientific Production and Most Cited Countries

Table 13 shows the affiliation countries’ frequency distribution. To understand the research productivity of a nation, the resulting frequency of scientific production by country was compared with the international monetary indicator Gross Domestic Product (GDP) 2019 World Bank [48]. The Gross Domestic Product (GDP) measures the monetary value of a country’s production of final goods and services during a year. GDP is commonly used as a measure of the degree of the well-being of a country’s population [49].

The USA was positioned in 1st place in scientific production with a frequency of 111, followed by GERMANY with a frequency of 27 and the UK also with the same frequency. These countries, apart from having the largest scientific production in H-Classics in consumer neuroscience, neuromarketing and neuroaesthetics, also occupy the first positions in the economic and welfare development international indexes (Figure 6 and Table 13).

Table 14 shows the countries that have received the highest total citations among the 50 H-Classics documents published and classified. Among the 50 selected papers, many of the selected authors chose articles to reference their scientific works having their origin of scientific production in the USA (2265 citations), followed by Germany (446 citations), Canada (406), United Kingdom (391 citations) and France (250 citations).

### 3.8. Sources, Countries and Keywords: Three-Fields Plot

Figure 7 shows relationships among Top Sources (left field), Top Countries (middle field), and Top Keywords (right field) and several items (1–10) summarized by a Sankey Plot. Sankey’s diagrams show the flows and their quantities in proportion to each other. The width of the arrows or lines is used to show their magnitudes, so the larger the arrow, the greater the amount of flow. The flow arrows or lines can be combined or divided through their paths at each stage of a process. The colour can be used to divide the diagram into different categories or to show the transition from one state of the process to another [51].

The USA was the country that published the H-Classics selection in the most relevant sources (Expert Systems with Applications, Nature Reviews Neuroscience, Journal of Cognitive Neuroscience, Journal of Marketing Research, Journal of Consumer Psychology, Psychology & Marketing, Frontiers in Human Neuroscience, Trends in Cognitive Sciences and Neuroimage. It was also the only country that used the Top 10 most frequently used keywords in the different studies (‘neuromarketing’, ‘neuroeconomics’, ‘neuroscience’, ‘consumer neuroscience’, ‘cognitive neuroscience’, ‘fMRI’, ‘EEG’, ‘neuroaesthetics’, ‘aesthetics’ and ‘music’). Out of the Top 10 countries, neither Spain nor Austria published in the Top main journals on the studied subject. The two keywords used in most of the countries were ‘fMRI’ and ‘neuroaesthetics’.

### 3.9. Most Global Cited Documents and Most Local Cited References

Table 15 shows the list of the most relevant manuscripts sorted by citations from the H-Classics selection (50 documents). Global Citations (TC) means the Total Citations that an article, included in the selected collection, has received from documents indexed on a bibliographic database (WoS, Scopus, etc.). So, TC counts citations received by a selected article “all over the world”. The most frequently cited work was ‘*Neuromarketing: the hope and hype of neuroimaging in business*’ [52], a research study about the application of neuroimaging methods to product marketing with 332 citations and a total citation per year of 30.18.

Table 16 shows the generated frequency table of the Most Cited References. It refers to the scientific document included in at least one of the reference lists (bibliography) of the document set. In our collection, we have more than 3700 references included in the 50 document bibliographies.

The most frequently cited local reference was the scientific work *‘Neural correlates of behavioural preference for culturally familiar drinks’* [60]. a research study where the authors delivered Coke and Pepsi to human subjects and examined them in behavioural taste tests and passive experiments carried out during functional magnetic resonance imaging (fMRI). The scientific work was cited 17 times in our H-Classics collection.

### 3.10. Conceptual Structure (2010–2019): Period View and Strategic Diagram (Network and Performance Measures Based on Words Analysis)

Figure 8A shows the Callon’s density and centrality as network measures [21,69] to each detected cluster in the selected period. The strategic diagram is divided into 4 quadrants (upper-right quadrant defines motor clusters, upper-left quadrant defines highly, developed, and isolated clusters, lower-left quadrant defines, emerging or declining clusters and lower-right quadrant defines basic and transversal clusters). Callon’s centrality measures the degree of interaction of a network with other networks, and it can be understood as the external cohesion of the network and Callon’s density measures the internal strength of the network, and it can be understood as the internal cohesion of the network. Figure 8B represents the strategic diagram during the period 2010–2019. Figure 8C shows the Quadrant distribution/Themes/Documents count. Figure 8D shows ART cluster, (E) PREFRONTAL CORTEX cluster, (F) NEUROMARKETING cluster, and (G) EEG cluster. Were analyzed the two quadrants (upper-right and lower-left) that we considered essential and most interesting for the development of the research area:

Upper-right or (Q1) Motor clusters with ART cluster (PREFERENCE, FACIAL ATTRACTIVENESS, NEUROPSYCHOLOGY, APPRECIATION, MUSIC CORRELATE, EMPIRICAL AESTHETICS, AESTHETICS, NEUROAESTHETICS, JUDGEMENT, PERCEPTION, and REWARDS VALUE) presents studies centred in experiments on aesthetic experiences by investigating behavioural, neural, and psychological properties of package design through fMRI, neuroimaging studies of positive-valence aesthetic appraisal across four sensory modalities, and studies related to systematic and literature reviews about neuroaesthetics and aesthetic experience.

Upper-right or (Q1) Motor cluster with NEUROMARKETING cluster, limiting with (Q2) Basic and Transversal cluster (ATTENTION, CONSUMER CHOICE, ADVERTISING, ETHICS, VALUE, ORBITOFRONTAL CORTEX, FMRI, REWARD, CONSUMER NEUROSCIENCE, VENTROMEDIAL PREFRONTAL CORTEX, and EMOTION) presents studies associated to an overview of the current and previous research in consumer psychology of brands and the role of researchers and practitioners when applying neuroscience to the consumer psychology of brands, investigations about physiological decision processes while participants undertook a choice task designed to elicit preferences for a product using EEG and eye-tracking.

Upper-right or (Q1) Motor cluster with PREFRONTAL CORTEX cluster, limiting with (Q4) Highly Developed and Isolated cluster (BRAIN ACTIVITY, AFFECTIVE STYLE, DECISION MAKING, and HIGH-RESOLUTION EEG) presents research focused on Applications, Challenges, and Possible Solutions in consumer neuroscience, changes in the EEG frontal activity during the observation of commercial videoclips, overviews of published papers about marketing research employing electroencephalogram (EEG) and magnetoencephalogram (MEG) methods and, fMRI studies using concealed information paradigm in which participants were trained to use countermeasures and to defeat deception detection.

Lower-left or Emerging or declining clusters (Q3) with EEG cluster (Marketing Research, Inhibition, Memory, Neural Responses, and Cortex) presents studies related to systematic and literature reviews about consumer neuroscience and marketing, brain responses to movie trailers to predict individual preferences for movies and commercial success, studies about abstract art and cortical motor activation through EEG, studies to predict consumers’ future choices through EEG and studies about the identification of frontal cortex activation in reaction to TV advertisements.

## 4. Discussion

In the present study, consumer neuroscience, neuromarketing and neuroaesthetics HCPs have been identified and consequently analyzed using the concept of H-Classics. The analysis of the HCPs allows us to highlight the following remarkable findings:

### 4.1. Leaders and Knowledge Hubs: Most Relevant Authors, Sources, Affiliations, and Countries

The 50 H-Classics documents analyzed are published in a total of 36 different sources. A total of 3727 references are used to illustrate the papers, which is an average of 74.54 references per H-Classics article. The document types are 39 articles, two proceeding papers, eight reviews, and one book chapter. It is important to note that among the 50 HCPs, 16% of them are literature reviews. Consequently, authors from the fields of consumer neuroscience, neuromarketing and neuroaesthetics use these H-Classics to reference their work in the theoretical framework of their papers. The H-Classics retrieved a sum of 282 keywords in the different papers selected, which means a sum of 5.64% of average keywords per document. The H-Classics in consumer neuroscience, neuromarketing and neuroaesthetics is written by a total of 210 authors. There is an average of 4.2% of authors per document, which means that the 50 HCPs collection has been written mostly by teams of four researchers/scientists. Only eight works out of 50 in total are written by a single author, this shows that in these disciplines there is a tendency to collaborate and co-produce with other authors.The distribution of H-Classics consists of publications from 2010–2017 even though the work is carried out from 2010 to 2019. 2011 is the most productive year in H-Classics with a total of 12 HCPs published. 2010 is the year with the highest average number of citations per year with seven H-Classics and a percentage of 115.14 citations/paper. Understandably, no work is produced in 2018–2019 that has entered the HCP selection due to the short period (24 months) for its full citation and comparison with other previously published work. Little time has passed since its publication to achieve the standards of citation of the H-Classics, for this reason, there are no studies yielded in that period. However, general scientific production in consumer neuroscience, neuromarketing and neuroaesthetics has not declined, with 142 publications in 2018 and 134 publications in 2019, respectively.The most relevant authors with the highest impact are scientist and professor Chatterjee (four articles published and with a total of 438 citations) and professor Nadal (also with four articles published and a total of 272 citations). Chatterjee is a Professor of Neurology at the Perelman School of Medicine at the University of Pennsylvania (USA). His research focuses on spatial cognition and its relationship to language. He also conducts neuroaesthetics research and writes about the ethical use of neuroscience findings in society. Besides, Chatterjee is the author of major references for the other authors of H-Classics, appearing cited 54 times, and is the author who, besides having one of the longest and most consistent trajectories in H-Classics during the period studied, has had H-Classics publications almost every year. The duration of global presence in a field of research/science should be considered as an indicator. For example, Chatterjee wrote his first publication (according to Google Scholar) in 1991 and has one citation (understanding this lower impact about HCPs, where everyone needs some time before being able to write high impact articles). Based on this reasoning, the year of an author’s first publication could logically also be chosen as a theoretically relevant parameter to evaluate the trajectory of HCPs. Nadal is a Professor in the Department of Psychology of the Faculties of Education, Nursing and Physiotherapy and Psychology of the Universitat de les Illes Balears (Spain). Nadal is a permanent staff of the Human Evolution and Cognition Research Group (EVOCOG). His research is focused on psychological aesthetics, neuroaesthetics, and the evolution of the mind. Venkatraman is the author, who, despite having one of the shortest trajectories in H-Classics during the studied period, presents several works in 2015, which makes him the author with the highest TC/TCpY (two H-Classics publications freq., 229 citations and total citations per year of 38,167).The most relevant source is Wiley’s *Journal of Consumer Psychology* (six H-Classics published and a total of 739 citations). Edited by Dr Anirban Mukhopadhyay (Hong Kong University of Science and Technology), The *Journal of Consumer Psychology* (JCP) publishes top-quality papers that contribute both theoretically and empirically to the understanding of the psychology of consumer behaviour. JCP is the official journal of the Society for Consumer Psychology and has a 5-year Impact factor of 5.140 and a 2019 Impact factor of 2.958, and the position it holds in the ISI Journal Citation Reports © Ranking: 2019 is 63/152 (Business) and 19/84 (Psychology, Applied). These results are to be expected since it is a journal with a long academic trajectory since it has been publishing articles on consumer behaviour and neuroscience since the 1990s, making it a reference within the scientific community. Also, the source growth indicates an exponential growth in the number of articles per selected publication, this means that during the first five years of the period (2010–2014) a slight number of HCPs has been published in the showed journals and it has been in the second part of the period when the journals have received the highest H-Classics volume. As we can see in the results related to the most cited authors and the most relevant journals, we observe that the most relevant authors are from the field of neuroesthetics; however, the journals where most HCPs articles are published are in consumer neuroscience, marketing, and commerce journals. Perhaps, at first glance it may seem that they are unconnected and separate disciplines, as pointed out by several authors, however, this fact could be a reason and reason for existing connections between the different subfields of neuroscience, and we would not have to consider them separately and watertight.Temple University (USA) is the university that has had the highest number of affiliations in H-Classics works in neuromarketing and neuroesthetics (a total of 12 articles). It is a benchmark institution in neuroscience and hosts the Center for Neural Decision Making, the Cognitive Neuroscience Laboratory, and Temple University Neurocognition Laboratory. Also, it should be noted that 60% of the most relevant institutions are from the USA. All of them have active research in neuromarketing, neuroaesthetics, or related fields, like consumer behaviour, neuroeconomics, and decision science. These results have shown that Temple’s scientists are at the forefront of research and teaching in the rapidly expanding world of neuroscience since they support an interdisciplinary approach to this exciting field of study, with different neuroscience programs spanning multiple Schools, Colleges, and research centres. However, Temple University occupies a relative position within the international rankings (301–400 in ARWU and 651–700 in QS). This means that, despite not being one of the most prestigious universities nor a university of generic excellence based in ARWU 2019 and QS 2019, it is one of the most relevant universities in this study discipline and a pioneer in research in neuromarketing and consumer neuroscience.Defining authorship in scientific articles and papers is an essential and complex process that involves subjectivity and depends on agreements generally established by word of mouth, which can lead to conflicts among researchers. The meaning of the order of the signature varies according to the areas. For example, in Mathematics, the order of the signatures is limited to an alphabetical criterion, therefore, the attribution of the value of the work is distributed equally among each author, and mathematicians are very careful not to collaborate with other researchers unless it is essential. Social Sciences has adopted the uses of Biomedicine where the order of authors implies different roles and workloads in the development of the article. Therefore, in this system, the positions reflect the role of each of the authors and the order the involvement in the work [70]. According to Robinson-García and Amat, we can identify these different types of authorship, such as first author, corresponding author, other authors and occasional collaborators [71]. For example, it seems that in some labs/countries the supervisor (i.e., the senior author, often the last author/head of the lab) is designated as the corresponding author rather than the first author (e.g., the PhD student who has done most of the work). These facts may to some extent distort or bias the perspective when analyzing some results, as the cultural as well as the social context must be considered to have a 360° perspective to interpret the results in an unbiased way. Within this context, we found that the USA is the country that has brought together the highest number of corresponding authors (19 articles). Besides, it is the country where most of its scientific production has been carried out through intra-collaboration. However, other countries like Canada, France, Spain, or Australia do not have intra-collaboration of H-Classics for the disciplines of consumer neuroscience, neuromarketing and neuroaesthetics and all the work they have produced has been through inter-collaboration. This can often be subject to conflicts of interest arising from the processing of sensitive data, as well as commercial interests, or permits to carry out neuroscience studies with humans, among others, making transnational research work difficult. The USA also has the highest distributed frequency of publications in H-Classics in neuromarketing and neuroesthetics (Freq. 11), is the country of origin of the references that more authors have cited in their bibliographies and is the country that has published in all the top 10 most relevant sources and used the top 10 most frequently keywords. The high level of development and quality of life expressed in GDP 2019/IMF 2019 can be reflected in the impact and high-quality scientific results and scientific production volume.

### 4.2. Disruptive Documents: Most Relevant Cited Papers, References, and Sources (from Reference List)

The most cited HCP is *‘Neuromarketing: the hope and hype of neuroimaging in business’* [52], a research study about the application of neuroimaging methods to product marketing with 332 citations and a total citation per year of 30.18. This work is a reference for all product marketers as it is an article review within the field of neuroimaging and its applicability in the field of business development.The most cited reference is *‘Neural correlates of behavioural preference for culturally familiar drinks’* [60], a research study where the authors delivered Coke and Pepsi to human subjects and examined their attitude and reactions in behavioural taste tests as well as in passive experiments carried out during functional magnetic resonance imaging (fMRI) to understand how cultural messages combine with content to shape our perceptions; even to the point of modifying behavioural preferences for a primary reward like a sugared drink. This scientific work was cited 17 times in our H-Classics collection. This work involves two giants of consumer goods and behind this type of studies always tends to have a large funding and a derived interest because it determines consumer behaviour in the global market, so they are works that are often disseminated in the news media and commonly the disclosure and impact are very high.The most cited source (from the reference list) is *Neuroimage* by Elsevier (182 referenced articles). Edited by Dr Michael Breakspear (The University of Sydney), *NeuroImage, a Journal of Brain Function*, provides a vehicle for communicating important advances in the use of neuroimaging to study structure-function and brain-behaviour relationships. The journal has a 5-year Impact factor: 6.682 and 2019 Impact factor: 5.902, and the position it occupies in the ISI Journal Citation Reports © Ranking: 2019 is 1/14 (Neuroimaging), 8/133 (Radiology, Nuclear Medicine) & Medical Imaging) and 33/271 (Neurosciences). This journal supposes a basic reference material since many consumer neuroscientists and neuroaesthetics experts use sources of this type to reference their work. This journal has a wide range of papers focused on consumer behaviour and consumer psychology by using the latest neuroscience techniques.

### 4.3. Conceptual Structure: Motor Themes, and Emerging or Declining Themes

The results of the co-word analysis reveal how motor themes in this discipline are focused on reviews of the literature on consumer neuroscience, neuroaesthetics and neuromarketing, the psychology of brand perception, and the role of researchers when applying neuroscience to brand design. It has been observed that there are motor studies focused on experiments related to the appreciation of aesthetics and visual and packaging design. The results show that research on the perception of decision making through neuroscience techniques, as well as work to detect deception, is a recurrent and basic topic. It is worth mentioning that among all the studied work, the 3 neuroscience main techniques revealed in emerging or declining themes are: EEG, fMRI, and eye-tracking, and most of the emerging work is focused on the clinical study on individuals who visualize TV advertisements, commercial videos or film trailers, and their consumer choices. The loss of effectiveness of advertising forces marketers to seek new tools to help them better understand the processing of information and consumer behaviour, for this reason, perhaps many of the works are focused on audiovisual content [72], since it may be more interesting to capture the emotion and feelings of the viewer and pay attention to the various stimuli that occur in different contexts during animated sequences.

## 5. Conclusions

This paper has allowed us to present the evolution of neuromarketing, neuroaesthetics, and consumer neuroscience during the last decade. Through the analysis of 50 HCPs, we highlighted the most remarkable authors, institutions, sources, and countries, as well as the most relevant documents, references, and driving forces or emerging or declining issues. The applicability and practicality of the present study should be remarked since it is a sample of information that is relevant to help understand and identify the academic network, composition, and structure of the past and present in the field of neuromarketing, and neuroaesthetics. This work acts as a frame of reference or a common indicator for people interested in the field of neuroesthetics, consumer neuroscience and neuromarketing when it comes to know with whom to research with, where to research, where to publish, which groups to collaborate with or what trends to research on, among others. This work enables new future research lines such as studying the impact of neuromarketing, and neuroaesthetics research development through other bibliometric indicators, measuring national and international scientific collaboration as well as future work centred within the current scientific production through the analysis of altmetrics [73] to evaluate the impact of the research work in the digital media ecosystems.

## Figures and Tables

**Figure 1 brainsci-11-00548-f001:**
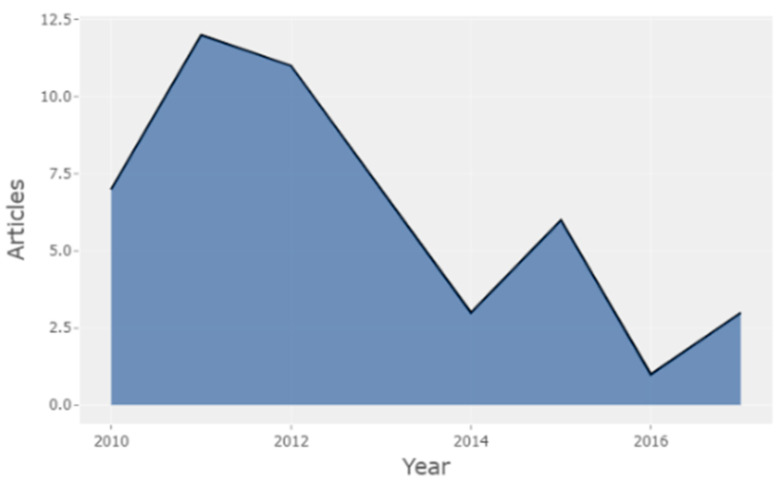
Distribution of publications by year and record count.

**Figure 2 brainsci-11-00548-f002:**
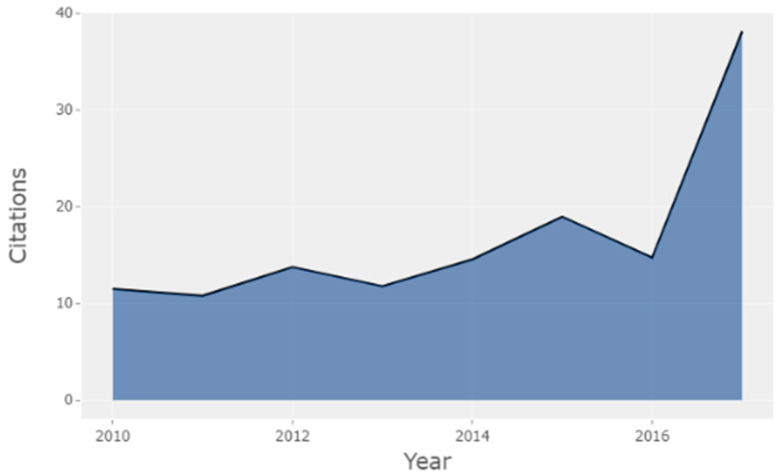
Total Citations per Year (TCpY).

**Figure 3 brainsci-11-00548-f003:**
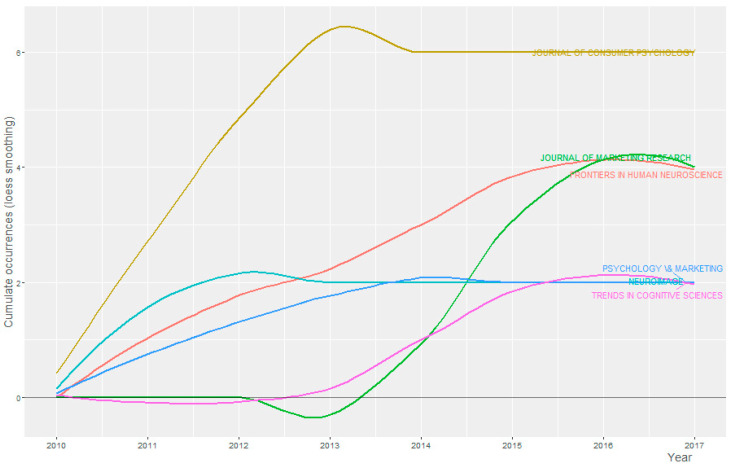
Source Growth. Cumulate Occurrences per Year and Number of Sources (5).

**Figure 4 brainsci-11-00548-f004:**
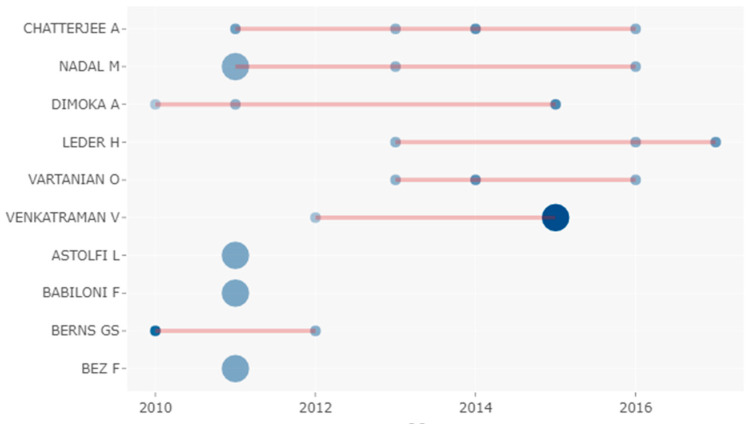
Top 10 Authors’ Production Over the Time Data Visualization. The line represents the author’s timeline. A bubble at a given year means that “XY” published at least a document in that year and the bubble size is proportional to the *n*. of documents XY published in that year. The colour intensity is proportional to the total citations per year of the document published in that year.

**Figure 5 brainsci-11-00548-f005:**
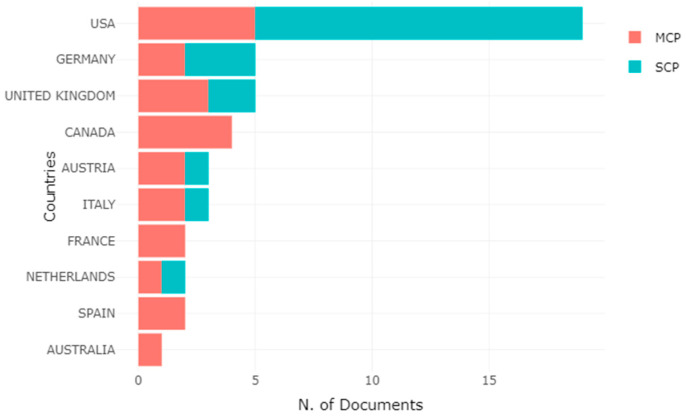
Top 10 Corresponding Author’s Country Data Visualization. Collaboration indices: The intra-country (SCP) and inter-country (MCP).

**Figure 6 brainsci-11-00548-f006:**
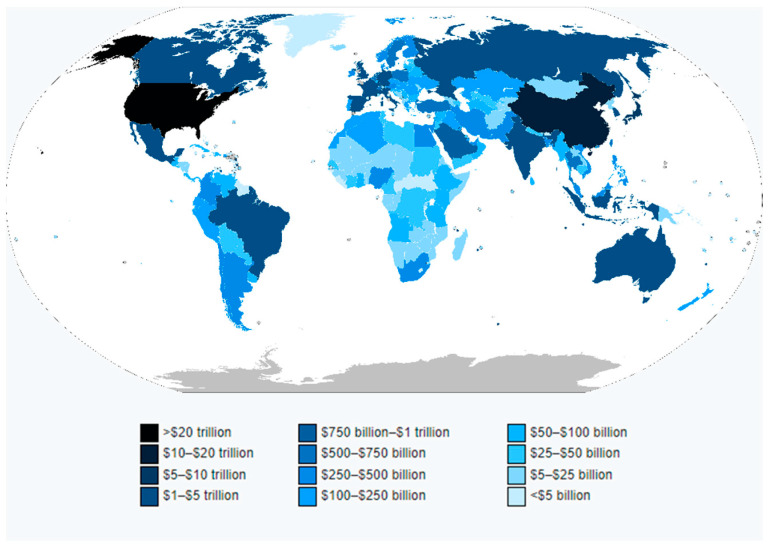
Countries by nominal GDP in 2019. Source: World Economic Outlook Database. International Monetary Fund (IMF). Retrieved: 13 October 2020 [50].

**Figure 7 brainsci-11-00548-f007:**
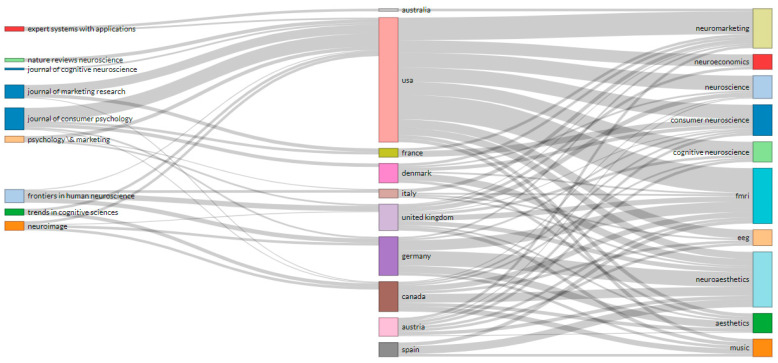
Three-Fields Plot. Middle Field (Countries): Number of items (1–10), Left Field (Sources): Number of items (1–10) and Right Field (Keywords): Number of items (1–10). Max width possible (1–50).

**Figure 8 brainsci-11-00548-f008:**
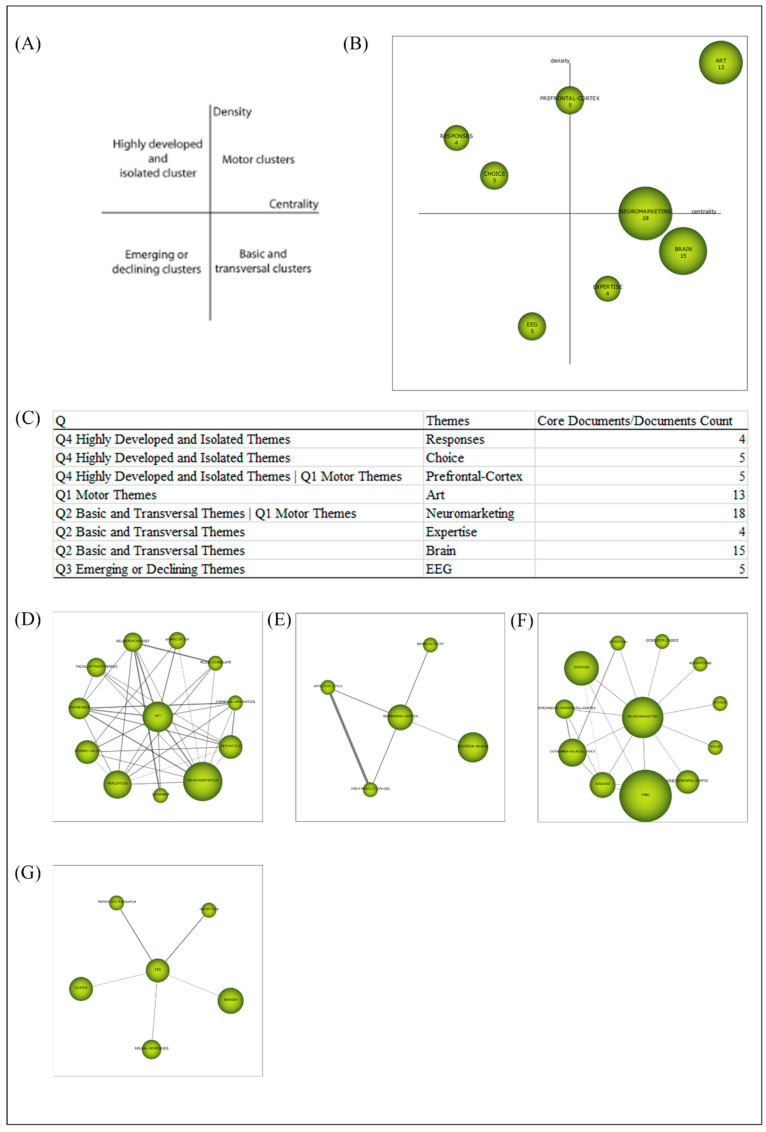
(**A**) Callon’s density and centrality measures. (**B**) SciMAT Strategic Diagram (2010–2019). (**C**) Quadrants/Themes/CoreDocuments-Documents Count. (**D**) ART cluster, (**E**) PREFRONTAL CORTEX cluster, (**F**) NEUROMARKETING cluster, and (**G**) EEG cluster.

**Table 1 brainsci-11-00548-t001:** Querying from Clarivate Analytics WoS.

Indexes	Timespan	Search	Results	Search Date
Web of Science Core Collection: SCI-EXPANDED, SSCI, A&HCI, CPCI-S, CPCI-SSH, BKCI-S, BKCI-SSH, ESCI, CCR-EXPANDED, IC.	2010–2019	TS = (“Neuromarketing” OR “Consumer Neuroscience” OR (“Neuroscience” AND (“Marketing” OR “Advert*”)) OR (“Neuroaesthetics” OR “Neuroesthetics”))	907	22 October 2020

**Table 2 brainsci-11-00548-t002:** Citation Report and Record Count.

Citation Report	Record Count
Results found	907
Sum of the Times Cited	9931
Average Citations per Item	10.95
H-Index	50

**Table 3 brainsci-11-00548-t003:** Main information about data.

Description	HCPs Results
Main Information about Data	
Timespan: There are no HCPs gathered during the last two years of the study period (2018–2019), however, the study period is 2010–2019.	2010:2017
Sources (Journals, Books, etc.)	36
Documents	50
Average Years from Publication (average years to an article to be cited)	7.52
Average Citations Per Documents	99.9
Average Citations Per Year Per Doc	12.53
References (Total number of documents cited in the HCPs collection)	3727
Document Types	
Article published in a Journal	39
Article published as a Proceedings Paper	2
A review published in a Journal	8
A review published as a Book Chapter	1
Document Contents	
Keywords Plus (Id): extracted from the titles of the cited references by Thomson Reuters.	282
Author’s Keywords (De): provided by the original Authors.	179
Authors	
Number of authors	210
The number of author appearances (while for the “authors per article” an author, even if he has published more than one article, is counted only once).	243
Authors of Single-Authored Documents	8
Authors of Multi-Authored Documents	202
Authors Collaboration	
Single-Authored Documents	8
Documents Per Author	0.238
Authors Per Document	4.2
Co-Authors Per Documents	4.86
Collaboration Index (CI): is calculated as Total Authors of Multi-Authored Articles/Total Multi-Authored Articles.	4.81

**Table 4 brainsci-11-00548-t004:** Average Citations per Year.

Year	Number of Articles	Total Citations per Article (TCpA)	Total Citations per Year (TCpY)	Citable Years (CY)
2010	7	115.1428571	11.51428571	10
2011	12	97.25	10.80555556	9
2012	11	110.0909091	13.76136364	8
2013	7	82.57142857	11.79591837	7
2014	3	87.33333333	14.55555556	6
2015	6	94.83333333	18.96666667	5
2016	1	59	14.75	4
2017	3	114.3333333	38.11111111	3

**Table 5 brainsci-11-00548-t005:** Top 10 Most Local Cited Sources (from Reference Lists).

Rank	Sources	Articles
1.	*Neuroimage*	182
2.	*Journal of Neuroscience*	117
3.	*Neuron*	111
4.	*Journal of Consumer Research*	99
5.	*Science*	87
6.	*PNAS*	82
7.	*Trends in Cognitive Sciences*	79
8.	*Nature Neuroscience*	67
9.	*Nature Reviews Neuroscience*	64
10.	*Journal of Consumer Psychology*	55

**Table 6 brainsci-11-00548-t006:** Top 10 Most Relevant Sources and Source Local Impact.

Rank	Source	Articles	H_Index	G_Index	M_Index	Total Citations (TC)	Number of Publications (NP)	Publication Year Start
1.	*Journal of Consumer Psychology*	6	6	6	0.545454545	739	6	2010
2.	*Frontiers in Human Neuroscience*	4	4	4	0.4	304	4	2011
3.	*Journal of Marketing Research*	4	4	4	0.666666667	389	4	2015
4.	*Neuroimage*	2	2	2	0.2	238	2	2011
5.	*Psychology & Marketing*	2	2	2	0.2	259	2	2011
6.	*Trends in Cognitive Sciences*	2	2	2	0.285714286	227	2	2014
7.	*Acta Psychologica*	1	1	1	0.111111111	128	1	2012
8.	*Annual Review of Psychology, Vol 63*	1	1	1	0.111111111	54	1	2012
9.	*Archives of General Psychiatry*	1	1	1	0.111111111	113	1	2012
10.	*Biosocieties*	1	1	1	0.25	58	1	2017

**Table 7 brainsci-11-00548-t007:** Source Dynamics.

		Highly Cited Papers (HCP) Published/Year							
Rank	Source	2010	2011	2012	2013	2014	2015	2016	2017
1.	*Journal of Consumer Psychology*	1	1	6	6	6	6	6	6
2.	*Neuroimage*	0	2	2	2	2	2	2	2
3.	*Psychology & Marketing*	0	1	1	2	2	2	2	2
4.	*Trends in Cognitive Sciences*	0	0	0	0	1	2	2	2
5.	*Journal of Marketing Research*	0	0	0	0	0	4	4	4
6.	*Frontiers in Human Neuroscience*	0	1	2	2	3	4	4	4

**Table 8 brainsci-11-00548-t008:** Top 10 Most Relevant Authors and Author Local Impact.

Rank	Author	Articles	H_Index	G_Index	M_Index	Total Citations (TC)	Number of Publications (NP)	Publication Year Start
1.	Chatterjee, A	4	4	4	0.4	438	4	2011
2.	Nadal, M	4	4	4	0.4	272	4	2011
3.	Dimoka, A	3	3	3	0.273	289	3	2010
4.	Leder, H	3	3	3	0.375	216	3	2013
5.	Vartanian, O	3	3	3	0.375	283	3	2013
6.	Venkatraman, V	3	3	3	0.333	289	3	2012
7.	Astolfi, L	2	2	2	0.2	141	2	2011
8.	Babiloni, F	2	2	2	0.2	141	2	2011
9.	Berns, G.S.	2	2	2	0.182	442	2	2010
10.	Bez, F	2	2	2	0.2	141	2	2011

**Table 9 brainsci-11-00548-t009:** Top 10 Most Local Cited Authors.

Rank	Authors	Citations
1	Chatterjee, A	54
2	Jacobsen, T	36
3	Knutson, B	30
4	Plassmann, H	30
5	Leder, H	24
6	Vartanian, O	23
7	Berridge, K.C.	22
8	Mcclure, S.M	22
9	Koelsch, S	21
10	Poldrack, R.A.	20

**Table 10 brainsci-11-00548-t010:** Top 10 Authors’ Production Over the Time.

Author	Year	Frequency	Total Citations (TC)	Total Citations per Year (TCpY)
Astolfi, L	2011	2	141	14.1
Babiloni, F	2011	2	141	14.1
Berns, G.S.	2010	1	332	30.182
Berns, G.S.	2012	1	110	12.222
Bez, F	2011	2	141	14.1
Chatterjee, A	2011	1	155	15.5
Chatterjee, A	2013	1	86	10.75
Chatterjee, A	2014	1	138	19.714
Chatterjee, A	2016	1	59	11.8
Dimoka, A	2010	1	57	5.182
Dimoka, A	2011	1	102	10.2
Dimoka, A	2015	1	130	21.667
Leder, H	2013	1	86	10.75
Leder, H	2016	1	59	11.8
Leder, H	2017	1	71	17.75
Nadal, M	2011	2	127	12.7
Nadal, M	2013	1	86	10.75
Nadal, M	2016	1	59	11.8
Vartanian, O	2013	1	86	10.75
Vartanian, O	2014	1	138	19.714
Vartanian, O	2016	1	59	11.8
Venkatraman, V	2012	1	60	6.667
Venkatraman, V	2015	2	229	38.167

**Table 11 brainsci-11-00548-t011:** Top 10 Corresponding Author’s Country.

Rank	Country	Articles	Frequency	Single-Country Publications (SCP)	Multiple-Country Publications (MCP)
1	USA	19	38	14	5
2	Germany	5	10	3	2
3	United Kingdom	5	10	2	3
4	Canada	4	8	0	4
5	Austria	3	6	1	2
6	Italy	3	6	1	2
7	France	2	4	0	2
8	The Netherlands	2	4	1	1
9	Spain	2	4	0	2
10	Australia	1	2	0	1

**Table 12 brainsci-11-00548-t012:** Top 10 Most Relevant Affiliations.

Rank	Organization	Country	Articles	ARWU 2019	QS 2019
1	Temple University	USA	12	301–400	651–700
2	University of Pennsylvania	USA	10	17	19
3	Duke University	USA	8	28	26
4	Freie Universität Berlin	Germany	8	-	130
5	Copenhagen Business School	Denmark	5	701–800	-
6	Emory University	USA	5	101–150	148
7	University of California, Los Angeles	USA	5	11	32
8	University of Oxford	UK	5	7	5
9	Columbia University	USA	4	8	16
10	Erasmus Universiteit Rotterdam	The Netherlands	4	68	179

Abbreviations: World University Rankings and 2019 Academic Ranking of World Universities (ARWU), Quacquarelli Symonds (QS).

**Table 13 brainsci-11-00548-t013:** Top 10 Country Scientific Production Data Visualization.

Rank	Country	Frequency	2019 Gross Domestic Product in GDP Nominal Rank	2019 Gross Domestic Product in GDP Nominal in Millions of US Dollars (2019 GDP/$)
1.	USA	111	#1	21,427,700
2.	Germany	27	#4	3,845,630
3.	UK	27	#6	2,827,113
4.	Canada	21	#10	1,736,426
5.	Italy	13	#8	2,001,244
6.	Denmark	12	#37	348,078
7.	Austria	11	#27	446,315
8.	Spain	10	#13	1,394,116
9.	Australia	9	#14	1,392,681
10.	France	8	#7	2,715,518

**Table 14 brainsci-11-00548-t014:** Top 10 Most Cited Countries.

Rank	Country	Total Citations (TC)	Average Article Citations (AAC)
1	USA	2265	119.2
2	Germany	446	89.2
3	Canada	406	101.5
4	United Kingdom	391	78.2
5	France	250	125
6	Italy	220	73.3
7	Brazil	214	214
8	Austria	180	60
9	Australia	161	161
10	The Netherlands	151	75.5

**Table 15 brainsci-11-00548-t015:** Top 10 Most Global Cited Document.

Rank	Paper	DOI	Reference	Total Citations (TC)	Total Citations per Year (TCpY)
1	Ariely D, 2010, *Nat Rev Neurosci*	10.1038/nrn2795	[52]	332	30.18
2	Lopes At, 2017, *Pattern Recognit*	10.1016/j.patcog.2016.07.026	[53]	214	53.50
3	Schmitt B, 2012, *J Consum Psychol*	10.1016/j.jcps.2011.09.005	[54]	170	18.89
4	Reimann M, 2010, *J Consum Psychol*	10.1016/j.jcps.2010.06.009	[55]	165	15.00
5	Khushaba RN, 2013, *Expert Syst Appl*	10.1016/j.eswa.2012.12.095	[5]	161	20.12
6	Brown S, 2011, *Neuroimage*	10.1016/j.neuroimage.2011.06.012	[56]	161	16.10
7	Chatterjee A, 2011, *J Cogn Neurosci*	10.1162/jocn.2010.21457	[57]	155	15.50
8	Plassmann H, 2012, *J Consum Psychol*	10.1016/j.jcps.2011.11.010	[58]	151	16.78
9	Spence C, 2011, *Psychol Mark*	10.1002/mar.20392	[59]	140	14.00
10	Morin C, 2011, *Society*	10.1007/s12115-010-9408-1	[1]	139	13.90

**Table 16 brainsci-11-00548-t016:** Top 10 Most Local Cited References (Rank/Cited References/Reference/Citations).

Rank	Cited References	Reference	Citations
1	McClure SM, 2004, *Neuron*, V44, P379, DOI 10.1016/J.Neuron.2004.09.019	[60]	17
2	Knutson B, 2007, *Neuron*, V53, P147, DOI 10.1016/J.Neuron.2006.11.010	[61]	15
3	Kawabata H, 2004, *J Neurophysiol*, V91, P1699, DOI 10.1152/Jn.00696.2003	[62]	13
4	Poldrack RA, 2006, *Trends Cogn Sci*, V10, P59, DOI 10.1016/J.Tics.2005.12.004	[63]	13
5	Jacobsen T, 2006, *Neuroimage*, V29, P276, DOI10.1016/J.Neuroimage.2005.07.010	[64]	12
6	Leder H, 2004, *Brit J Psychol*, V95, P489, DOI 10.1348/0007126042369811	[65]	12
7	Vartanian O, 2004, *Neuroreport*, V15, P893, DOI 10.1097/00001756-200404090-00032	[66]	12
8	Ariely D, 2010, *Nat Rev Neurosci*, V11, P284, DOI 10.1038/Nrn2795	[52]	11
9	Blood Aj, 2001, *P Natl Acad Sci USA*, V98, P11818, DOI 10.1073/Pnas.191355898	[67]	11
10	Aharon I, 2001, *Neuron,* V32, P537, DOI 10.1016/S0896-6273(01)00491-3	[68]	10

## Data Availability

The data presented in this study are openly available in the Zenodo repository at https://doi.org/10.5281/zenodo.4684621 (accessed on 13 April 2021).

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
