# Peer review of "Citation Classics in Consumer Neuroscience, Neuromarketing and Neuroaesthetics: Identification and Conceptual Analysis"

_brainsci, 2021, doi:10.3390/brainsci11050548_

Round 1
Reviewer 1 Report
I read the article with great curiosity, and it is an overall very interesting approach to highlight the most important developments in the fields of consumer neuroscience and neuroaesthetics. It is easy to follow and well written.
There are only two minor things that I want to highlight as possible improvements:
First, the authors use the terms "neuromarketing" and "consumer neuroscience" quite indifferent. In recent years, it has become the standard that consumer neuroscience is the discipline that investigates the neural correlates of consumer decisions with a clear focus on progress in basic scientific understanding of the brain, whereas neuromarketing is the discispline that tries to use the methodologies and insights from consumer neuroscience in applied research and business applications. I think the manuscript would benefit if this distinction is explicitly referenced, since I get the feeling that oftentimes when the authors write about neuromarketing, they actually mean consumer neuroscience.
Secondly, it is a great achievement to identify the 50 most influential publications in that area. For the reader, it would however be very valuable to have a table listing all 50 publications, including a short summary of the respective studies. This way, this manuscript would be enriched with a short review and the reader gets introduced into the research area as a whole.
Author Response
I read the article with great curiosity, and it is an overall very interesting approach to highlight the most important developments in the fields of consumer neuroscience and neuroaesthetics. It is easy to follow and well written.
There are only two minor things that I want to highlight as possible improvements:
First, the authors use the terms "neuromarketing" and "consumer neuroscience" quite indifferent. In recent years, it has become the standard that consumer neuroscience is the discipline that investigates the neural correlates of consumer decisions with a clear focus on progress in basic scientific understanding of the brain, whereas neuromarketing is the discispline that tries to use the methodologies and insights from consumer neuroscience in applied research and business applications. I think the manuscript would benefit if this distinction is explicitly referenced, since I get the feeling that oftentimes when the authors write about neuromarketing, they actually mean consumer neuroscience.
Dear reviewer,
We are very grateful that you have reviewed the manuscript and very happy that you liked it.
We wrote a precise definition. Furthermore, we have included almost 90% of the definition identically that you provided as we believe it is a very professional explanation and very concise. We hope that now it is a little clearer and does not lead to confusion about the different terms.
Secondly, it is a great achievement to identify the 50 most influential publications in that area. For the reader, it would however be very valuable to have a table listing all 50 publications, including a short summary of the respective studies. This way, this manuscript would be enriched with a short review and the reader gets introduced into the research area as a whole.
We have included in open access the citation reports (with the 50HCPs) as well as the different datasets in case other readers or researchers want to go deeper into the subject. We have added it to the European repository Zenodo (you can check the link in materials and methods or at the end of the manuscript).
Thank you very much. Best regards

Reviewer 2 Report
“Citation Classics in Neuromarketing and Neuroaesthetics Identification and Conceptual Analysis”
To start, I’m not an expert in neuromarketing nor infometrics, but I accepted the invitation to review based on my expertise in neuroaesthetics and because the abstract sounded interesting. The main issue is that the theoretical reasoning behind the paper is unclear (Major points 1,2,4,5) therefore it becomes also unclear what the point of the paper is and what can be learned from the paper (major point 2 & 5) this makes the results a bit of a stream of tables where the reader cannot see the forest for the trees (major point 4-5) as well as that the theoretical underpinnings behind the analyses are often unclear (major point 4-5). Note that in lines 624-625 the authors seem to hint at what the relevance for their paper is but it remains unclear how this is achieved by the current paper. Finally, detail about the exact analyses are missing, in some cases this would be fine for a sophisticated reader in infometrics who could fill in the gaps themselves but it makes the paper less accessible to other readers which is a shame as I would assume that at least scholars from neuromarketing and neuroaesthetics (who may not have an extensive knowledge of infometrics). Below, I explain my points in more detail with the aim of helping the authors revise the paper for future submissions.
Major points:
-
- Perhaps, because of my background I do not necessarily think the link between neuromarketing and neuroaesthetics is so clear. Unfortunately, the reasoning as to why these fields were here considered simultaneously also did not because clear to me when reading the paper. For example, I would argue with the statement that neuroaesthetics is trying to find universal standards of beauty and to separate them from personal tastes or passing styles of fashion. As very often, it does deal with subjective ratings of beauty (or other evaluation questions, e.g. liking). As no references are given it is not clear to me what this is based on. Note that I am aware that people have argued for such a combination, specifically Skov and Nadal (2018,2020) nonetheless as these authors felt that it was worth following up their 2018 paper with a similar argumentation in 2020 I think it seems safe to say that their view is not necessarily widely shared by the research community, and that, though there may be good arguments to merge neuromarketing and neuroaesthetics, the fields currently are separated.
- In the same vein, at least reference 6 and 7 (lines 55-56) I do not think can be used to support the statement made in lines 52-56. Based on the abstract of 8 (I did not have time to read the paper in detail) the same seems to apply to 8.
- Perhaps, because of my background I do not necessarily think the link between neuromarketing and neuroaesthetics is so clear. Unfortunately, the reasoning as to why these fields were here considered simultaneously also did not because clear to me when reading the paper. For example, I would argue with the statement that neuroaesthetics is trying to find universal standards of beauty and to separate them from personal tastes or passing styles of fashion. As very often, it does deal with subjective ratings of beauty (or other evaluation questions, e.g. liking). As no references are given it is not clear to me what this is based on. Note that I am aware that people have argued for such a combination, specifically Skov and Nadal (2018,2020) nonetheless as these authors felt that it was worth following up their 2018 paper with a similar argumentation in 2020 I think it seems safe to say that their view is not necessarily widely shared by the research community, and that, though there may be good arguments to merge neuromarketing and neuroaesthetics, the fields currently are separated.
- Besides the combination of the two fields a more pressing point is that the theoretical reasoning in general is unclear to me. What is the interesting research question here? As it stands, the paper seems to not go beyond “we have a new cool analysis let’s try it out!”, though I understand this fun for the researchers to do, I do not currently see what knowledge is gained and, more importantly, what the informational value of this knowledge is, i.e. why is it interesting? What do we learn?
Unfortunately, this is not only not addressed in the introduction but also not in the discussion. Though the discussion promises some contextualization of the findings, it mainly reads as a repetition of the findings and I, as a reader basically had to come up myself with what the point and purpose of the paper was. - Missing information:
- Starting from the introduction, terms are not properly explained. E.g., no definition or description of the H-index or H-core is given. And this happens also throughout the results, e.g. “phase 2” of section 2.2. is to compute the h-index but the authors never give the calculation. As the h-index is well-known, many readers may be familiar with the computation, nonetheless it would always be useful for the authors to be specific. Furthermore, this happens also for computations that are less well-known or where in some cases it seems the authors have computed their own index. For example, it is not clear how the less well-known H-classics index is derived and from lines 142-143 this remains obscure. The same can be said for e.g. Total citations per year or citable years, M-index, G-index, etc.
Note that in some cases the reader can guess, e.g. Table 8 I assume that TC = total citations, NP= number of citations and PY_start =is the first year the author published in (in the set of included studies) but the reader shouldn’t guess, the authors should make this clear and unambiguous. Note here that here again it is needed to explain more than just what the ancronym stands for but also how things are calculated as the “PY_start” illustrates. As noted, I guess is now considered ONLY within the articles selected for the stimuli, as I’m sure that e.g. the number 1 and 2 (Chatterjee and Nadal) published papers before 2011. But of course, it would be possible to also consider their very first publication in general. Which may also be useful, as later briefly mentioned in the discussion these scientists are established professors, this would also be indicated by the length of their overall presence in a research field/science, for example, Chatterjee’s first publication (according to Google Scholar was in 1991 and has 1 citation (perhaps indicating that everyone needs some time before they can write highly impactful papers). Based on this reasoning, the year of the very first publication of an author could logically also be chosen as a theoretically relevant parameter.
- In some cases terms (e.g. inter vs intra collaboration) show up in the discussion but have not previously been defined or mentioned.
- There is also other missing information, generally in the tables (that could also do with reformatting, see minor point 3). For example, in Table 3, what does “References” refer to? Similarly it is unclear what is meant with keywords plus (ID0 and Author’s Keywords (De) and what the difference between the two is. Also what does “Article; Proceedings Paper” refer to, is it an article (for which there is a separate category) or a proceedings apper? Same for the Review;Book Chapter category. Furthermore what does “Author appearances” mean? Also the timespan is noted here as 2010-2017 whereas beforehand it was noted that the timespan was 2010-2019. Finally, “Average years from publication” reports a 7,52 average, which considering a 7 year time span seems confusing in addition to that it is not clear what is counted here, average years from publication to what? Until it is cited? Or what is meant here? Note that similar issues happen in other parts of the paper as well, e.g. no matter which time span, an article published in 2010 cannot have “citable years” of 10 as the maximum possible would be 9 using a timespan of 2010-2019? As it seems that though the paper can be cited already in 2010 this would not count as an extra year? In a way, this second example also underscores point 3a) I guess depending on how you count 10 might be the maximum but since it is unclear how citable years is calculated this remains unclear.
- Note that this also applies to figures. For example, Figure 7, here plotting parameters are missing that are needed to interpret the figure. E.g. what is the max width possible? Also, the figure is not compatible with table 5-6, specifically 2 journals (expert systems with applications, journal of cognitive neuroscience) are not listed in the table but are noted in the figure.
- A huge table of all included papers for the analysis probably would not help anyone, however, why not make the dataset used for the analysis publicly available through e.g. the OSF? Or at least make the list of all included publications available? This seems easily doable and would make the results of the authors replicable and controllable by the scientific community.
- The theoretical reasoning behind choices made in the method section stay unclear overall. To name a few specific examples:
- Why was a range of 2010-2019 chosen, if the attempt is to look at “classic” papers, would one not assume that these are often older papers that have “passed the test of time”?
- The search resulted in 907 hits, it seems that all papers were blindly included? It would make sense to check for applicability as I know from my experience with meta-analysis that one often finds search results that do not apply (e.g. “art” also stands for “artificial reproductive therapy” which can create havoc in search results). Perhaps, in this case all articles are suitable but it seems sensible to check for this.
- Why was a range of 2010-2019 chosen, if the attempt is to look at “classic” papers, would one not assume that these are often older papers that have “passed the test of time”?
- The theoretical reasoning behind why to perform certain analyses and their interpretation is unclear. Because of this, the reader is confronted with a “tsunami” of tables and numbers with no idea on what these tables/numbers should be communicating to the reader or how to interpret them. As it stands, tables and numbers are merely given but not contextualized. To name a few examples:
- In lines 185-193 the authors seem to disregard that papers published in 2019 hardly have the chance to be cited by papers published in 2019 and before. To be fair, the authors acknowledge this in the discussion but this result is provided without context which I think is rather exemplary for how the results overall are presented. Of course, and in-depth discussion of the results and how they connect to wider literature should be reserved for the discussion. However, some interpretation would be needed already in the results. Without this, it remains unclear what the results mean and what the reader should be getting from this.
- Why are the results collapsed over neuromarketing and neuroaesthetics? It seems to me to split the results up, see also point 1b). Note that this may also shed some light on the results, I noticed that in the results the top journals for example seem very marketing focused, but the two top authors (Chatterjee, nadal) generally would be more in neuroaesthetics rather than neuromarketing. Perhaps this could also be a reason to keep the fields together but since the authors leave these issues unmentioned it would in any case call for some more interpretation and contextualization.
- What is the theoretical relevance of e.g. section 3.4. why is this analysis theoretically relevant? What is it telling us? Note that as it stands this could apply to basically all sections. Similarly, why is institute of affiliation or country relevant? Note that in some cases, I can come up with explanations myself, but again this is not my job. The authors should know why they do certain analyses and what theoretical relevance is.
- What is the theoretical reasoning behind the choices concerning authorship.
Sometimes this is simply unclear (e.g. missing information), g. in Table 9 I assume that the 54 citations of Chatterjee are counted across all publications he was an author of rather than only first-author publications? But how co-authorship is dealt with is relevant especially since the authors find that the field generally includes co-authored papers. For example, based on Figure 4 it seems that Astolfi, Babiloni, and Bez wrote an article together that was published in 2011, however this connection seems to be never made by the authors. Similarly, why is the focus of page 12 on the corresponding author rather than the first author? I have noticed that different labs—and to some extent, different countries) seem to deal with the issue of the corresponding author differently. I.e. it seems practice in some labs/countries that the supervisor (i.e. the senior author, often the last author/ head of the lab) is designated as the corresponding author rather than the first author (e.g. the PhD student who carried out most of the work). Note that I do not mean here to only criticize this practice, e.g. in the case of master students (especially in systems that have separated masters and phd programmes) who are first author it is very likely that after the project they will leave academia and as many PhD students also leave academia after completion of their degree the same could be said for them. In this context, it may be fruitful to have a corresponding author (e.g. the senior author) who is still actually reachable and available for inquires about the paper after it has published rather than an unreachable author. Nonetheless, the choice of using corresponding author instead of a first author (or vice versa) needs to be theoretically explained as substantiated as this choice influences the results as well as the interpretation. - The use of the GDP baffles me, what is the research question here? Why is this relevant?
- In lines 185-193 the authors seem to disregard that papers published in 2019 hardly have the chance to be cited by papers published in 2019 and before. To be fair, the authors acknowledge this in the discussion but this result is provided without context which I think is rather exemplary for how the results overall are presented. Of course, and in-depth discussion of the results and how they connect to wider literature should be reserved for the discussion. However, some interpretation would be needed already in the results. Without this, it remains unclear what the results mean and what the reader should be getting from this.
Minor points:
- Lines 57-80, I found this section very vague and had to read it over several times until I had a feeling I grasped the authors meaning. Hence, it is probably worth revising this section.
- Section 2.2., because the formatting of the 4 key phases differs I first only identified 3 phases and was confused until I realized that lines 121-124 were meant to represent “phase 1”. I think here some simple formatting (e.g. just numbering) would make this section a lot clearer.
- In general, it’s worth rethinking the formatting of the tables, in many cases the formatting can be improved to create a table that is easier to visually understand and incorporates more information. Note that in some cases there seems to be some general issues such as table 7 where text overlaps/
- Figure 2, the peak seems to be in 2017 not 2010 as the text claims.
- Line 496 what is meant with “to an average of 4.2% of authors per scientific work”?
Author Response
“Citation Classics in Neuromarketing and Neuroaesthetics Identification and Conceptual Analysis”
To start, I’m not an expert in neuromarketing nor infometrics, but I accepted the invitation to review based on my expertise in neuroaesthetics and because the abstract sounded interesting. The main issue is that the theoretical reasoning behind the paper is unclear (Major points 1,2,4,5) therefore it becomes also unclear what the point of the paper is and what can be learned from the paper (major point 2 & 5) this makes the results a bit of a stream of tables where the reader cannot see the forest for the trees (major point 4-5) as well as that the theoretical underpinnings behind the analyses are often unclear (major point 4-5). Note that in lines 624-625 the authors seem to hint at what the relevance for their paper is but it remains unclear how this is achieved by the current paper. Finally, detail about the exact analyses are missing, in some cases this would be fine for a sophisticated reader in infometrics who could fill in the gaps themselves but it makes the paper less accessible to other readers which is a shame as I would assume that at least scholars from neuromarketing and neuroaesthetics (who may not have an extensive knowledge of infometrics). Below, I explain my points in more detail with the aim of helping the authors revise the paper for future submissions.
Dear reviewer, we are very grateful that you took the time to review the paper. We appreciate all the very constructive comments and hope we have corrected all the errors.
Major points:
-
- Perhaps, because of my background I do not necessarily think the link between neuromarketing and neuroaesthetics is so clear. Unfortunately, the reasoning as to why these fields were here considered simultaneously also did not because clear to me when reading the paper. For example, I would argue with the statement that neuroaesthetics is trying to find universal standards of beauty and to separate them from personal tastes or passing styles of fashion. As very often, it does deal with subjective ratings of beauty (or other evaluation questions, e.g. liking). As no references are given it is not clear to me what this is based on. Note that I am aware that people have argued for such a combination, specifically Skov and Nadal (2018,2020) nonetheless as these authors felt that it was worth following up their 2018 paper with a similar argumentation in 2020 I think it seems safe to say that their view is not necessarily widely shared by the research community, and that, though there may be good arguments to merge neuromarketing and neuroaesthetics, the fields currently are separated.
We replaced the definition. Your explanation help us to changed since we realized led to confusion. We have provided a more detailed definition of neuroesthetics (with ref.). and included the an explanation about the merge of neuromarketing and neuroaesthetics (Skov and Nadal).
- In the same vein, at least reference 6 and 7 (lines 55-56) I do not think can be used to support the statement made in lines 52-56. Based on the abstract of 8 (I did not have time to read the paper in detail) the same seems to apply to 8.
Thank you very much. We have replaced the references and we have included some that we believe support and support our text, in a more precise way. We have also modified the sentence to make it more meaningful.
- Perhaps, because of my background I do not necessarily think the link between neuromarketing and neuroaesthetics is so clear. Unfortunately, the reasoning as to why these fields were here considered simultaneously also did not because clear to me when reading the paper. For example, I would argue with the statement that neuroaesthetics is trying to find universal standards of beauty and to separate them from personal tastes or passing styles of fashion. As very often, it does deal with subjective ratings of beauty (or other evaluation questions, e.g. liking). As no references are given it is not clear to me what this is based on. Note that I am aware that people have argued for such a combination, specifically Skov and Nadal (2018,2020) nonetheless as these authors felt that it was worth following up their 2018 paper with a similar argumentation in 2020 I think it seems safe to say that their view is not necessarily widely shared by the research community, and that, though there may be good arguments to merge neuromarketing and neuroaesthetics, the fields currently are separated.
- Besides the combination of the two fields a more pressing point is that the theoretical reasoning in general is unclear to me. What is the interesting research question here? As it stands, the paper seems to not go beyond “we have a new cool analysis let’s try it out!”, though I understand this fun for the researchers to do, I do not currently see what knowledge is gained and, more importantly, what the informational value of this knowledge is, i.e. why is it interesting? What do we learn?
Unfortunately, this is not only not addressed in the introduction but also not in the discussion. Though the discussion promises some contextualization of the findings, it mainly reads as a repetition of the findings and I, as a reader basically had to come up myself with what the point and purpose of the paper was.
Thank you. We included in the introduction the purpose of the article and what we hope to answer with our manuscript. - Missing information:
- Starting from the introduction, terms are not properly explained. E.g., no definition or description of the H-index or H-core is given. And this happens also throughout the results, e.g. “phase 2” of section 2.2. is to compute the h-index but the authors never give the calculation. As the h-index is well-known, many readers may be familiar with the computation, nonetheless it would always be useful for the authors to be specific. Furthermore, this happens also for computations that are less well-known or where in some cases it seems the authors have computed their own index. For example, it is not clear how the less well-known H-classics index is derived and from lines 142-143 this remains obscure. The same can be said for e.g. Total citations per year or citable years, M-index, G-index, etc.
We have included a more extensive definition in in the introduction as well as in materials and methods, to make clearer the purpose and how to calculate both indexes. Also, we numbered the steps of the methodology.
Note that in some cases the reader can guess, e.g. Table 8 I assume that TC = total citations, NP= number of citations and PY_start =is the first year the author published in (in the set of included studies) but the reader shouldn’t guess, the authors should make this clear and unambiguous. Note here that here again it is needed to explain more than just what the ancronym stands for but also how things are calculated as the “PY_start” illustrates. As noted, I guess is now considered ONLY within the articles selected for the stimuli, as I’m sure that e.g. the number 1 and 2 (Chatterjee and Nadal) published papers before 2011. But of course, it would be possible to also consider their very first publication in general. Which may also be useful, as later briefly mentioned in the discussion these scientists are established professors, this would also be indicated by the length of their overall presence in a research field/science, for example, Chatterjee’s first publication (according to Google Scholar was in 1991 and has 1 citation (perhaps indicating that everyone needs some time before they can write highly impactful papers). Based on this reasoning, the year of the very first publication of an author could logically also be chosen as a theoretically relevant parameter.
We changed the tables and Improved the descriptive information. Also, we changed discussion with your valuable insights about the cited authors.
- In some cases terms (e.g. inter vs intra collaboration) show up in the discussion but have not previously been defined or mentioned.
We have included a definition in the description in the results as well as in the figure of Corresponding authors.
- There is also other missing information, generally in the tables (that could also do with reformatting, see minor point 3). For example, in Table 3, what does “References” refer to? Similarly it is unclear what is meant with keywords plus (ID0 and Author’s Keywords (De) and what the difference between the two is. Also what does “Article; Proceedings Paper” refer to, is it an article (for which there is a separate category) or a proceedings apper? Same for the Review;Book Chapter category. Furthermore what does “Author appearances” mean? Also the timespan is noted here as 2010-2017 whereas beforehand it was noted that the timespan was 2010-2019. Finally, “Average years from publication” reports a 7,52 average, which considering a 7 year time span seems confusing in addition to that it is not clear what is counted here, average years from publication to what? Until it is cited? Or what is meant here? Note that similar issues happen in other parts of the paper as well, e.g. no matter which time span, an article published in 2010 cannot have “citable years” of 10 as the maximum possible would be 9 using a timespan of 2010-2019? As it seems that though the paper can be cited already in 2010 this would not count as an extra year? In a way, this second example also underscores point 3a) I guess depending on how you count 10 might be the maximum but since it is unclear how citable years is calculated this remains unclear.
- No, it is possible.
All the required missing information was extended and included in the table.
If an article is published in 2010 could have 10 citable years till 2019 (for example, if we publish one paper in 2010, we have: 2010, 2011, 2012, 2013, 2014, 2015, 2016, 2017, 2018, 2019 to be cited, in total 10 years). So, its 10 years not 9. The publishing year of the paper is included, if it was published in 2010, and the same year got citations, also are counted. here are no HCPs gathered during the last two years of the study period (2018-2019), however the study period is 2010-2019. - Note that this also applies to figures. For example, Figure 7, here plotting parameters are missing that are needed to interpret the figure. E.g. what is the max width possible? Also, the figure is not compatible with table 5-6, specifically 2 journals (expert systems with applications, journal of cognitive neuroscience) are not listed in the table but are noted in the figure.
We included more information in the Figure description as well as the max width possible. The difference between the figure 7 and the table 5-6 is that in the figure 7 we are measuring the use of keywords in different countries and sources and in table 5-6 we are measuring the volume of scientific production (number of articles) not the frequency of keywords. I hope it is clearer the concept now. - A huge table of all included papers for the analysis probably would not help anyone, however, why not make the dataset used for the analysis publicly available through e.g. the OSF? Or at least make the list of all included publications available? This seems easily doable and would make the results of the authors replicable and controllable by the scientific community.The retrieved WoS dataset and the citation report are now openly available at Zenodo repository.
- The theoretical reasoning behind choices made in the method section stay unclear overall. To name a few specific examples:
- Why was a range of 2010-2019 chosen, if the attempt is to look at “classic” papers, would one not assume that these are often older papers that have “passed the test of time”?
A more precise definition was included in materials and methods. Regarding the citation classics, they are not old articles or articles with a long trajectory (antiquity). The concept of "classics" refers to articles that have become classics, i.e. what would be "bestsellers in the commercial literature". This consideration of classics is not associated with age but with the number of accumulated citations, which makes them high impact articles and a reference for other authors.
- The search resulted in 907 hits, it seems that all papers were blindly included? It would make sense to check for applicability as I know from my experience with meta-analysis that one often finds search results that do not apply (e.g. “art” also stands for “artificial reproductive therapy” which can create havoc in search results). Perhaps, in this case all articles are suitable but it seems sensible to check for this.
Were checked for applicability and described in materials and methods.
- Why was a range of 2010-2019 chosen, if the attempt is to look at “classic” papers, would one not assume that these are often older papers that have “passed the test of time”?
- The theoretical reasoning behind why to perform certain analyses and their interpretation is unclear. Because of this, the reader is confronted with a “tsunami” of tables and numbers with no idea on what these tables/numbers should be communicating to the reader or how to interpret them. As it stands, tables and numbers are merely given but not contextualized. To name a few examples:
- In lines 185-193 the authors seem to disregard that papers published in 2019 hardly have the chance to be cited by papers published in 2019 and before. To be fair, the authors acknowledge this in the discussion but this result is provided without context which I think is rather exemplary for how the results overall are presented. Of course, and in-depth discussion of the results and how they connect to wider literature should be reserved for the discussion. However, some interpretation would be needed already in the results. Without this, it remains unclear what the results mean and what the reader should be getting from this.
Of course, you are right. Thank you.
We included this appreciation in results. - Why are the results collapsed over neuromarketing and neuroaesthetics? It seems to me to split the results up, see also point 1b). Note that this may also shed some light on the results, I noticed that in the results the top journals for example seem very marketing focused, but the two top authors (Chatterjee, nadal) generally would be more in neuroaesthetics rather than neuromarketing. Perhaps this could also be a reason to keep the fields together but since the authors leave these issues unmentioned it would in any case call for some more interpretation and contextualization.
Thank you very much for this valuable point of view. We have included this perspective in the discussion, it is very interesting and even this relationship supports the results. - What is the theoretical relevance of e.g. section 3.4. why is this analysis theoretically relevant? What is it telling us? Note that as it stands this could apply to basically all sections. Similarly, why is institute of affiliation or country relevant? Note that in some cases, I can come up with explanations myself, but again this is not my job. The authors should know why they do certain analyses and what theoretical relevance is.
We hope to reflect the theoretically relevance in the changes made in the introduction as well as in the conclusion. - What is the theoretical reasoning behind the choices concerning authorship.
Sometimes this is simply unclear (e.g. missing information), g. in Table 9 I assume that the 54 citations of Chatterjee are counted across all publications he was an author of rather than only first-author publications? But how co-authorship is dealt with is relevant especially since the authors find that the field generally includes co-authored papers. For example, based on Figure 4 it seems that Astolfi, Babiloni, and Bez wrote an article together that was published in 2011, however this connection seems to be never made by the authors. Similarly, why is the focus of page 12 on the corresponding author rather than the first author? I have noticed that different labs—and to some extent, different countries) seem to deal with the issue of the corresponding author differently. I.e. it seems practice in some labs/countries that the supervisor (i.e. the senior author, often the last author/ head of the lab) is designated as the corresponding author rather than the first author (e.g. the PhD student who carried out most of the work). Note that I do not mean here to only criticize this practice, e.g. in the case of master students (especially in systems that have separated masters and phd programmes) who are first author it is very likely that after the project they will leave academia and as many PhD students also leave academia after completion of their degree the same could be said for them. In this context, it may be fruitful to have a corresponding author (e.g. the senior author) who is still actually reachable and available for inquires about the paper after it has published rather than an unreachable author. Nonetheless, the choice of using corresponding author instead of a first author (or vice versa) needs to be theoretically explained as substantiated as this choice influences the results as well as the interpretation.
We include a precise explanation in discussion. Thank you very much for your insights. - The use of the GDP baffles me, what is the research question here? Why is this relevant?
We compared the research productivity of the nations. We improved the description of the results with the aim to make it clearer and more understandable.
- In lines 185-193 the authors seem to disregard that papers published in 2019 hardly have the chance to be cited by papers published in 2019 and before. To be fair, the authors acknowledge this in the discussion but this result is provided without context which I think is rather exemplary for how the results overall are presented. Of course, and in-depth discussion of the results and how they connect to wider literature should be reserved for the discussion. However, some interpretation would be needed already in the results. Without this, it remains unclear what the results mean and what the reader should be getting from this.
Minor points:
- Lines 57-80, I found this section very vague and had to read it over several times until I had a feeling I grasped the authors meaning. Hence, it is probably worth revising this section.
With the expansion of the introduction and the points previously discussed, we believe that it is now better understood.
- Section 2.2., because the formatting of the 4 key phases differs I first only identified 3 phases and was confused until I realized that lines 121-124 were meant to represent “phase 1”. I think here some simple formatting (e.g. just numbering) would make this section a lot clearer.
We changed and included the 4 phases. - In general, it’s worth rethinking the formatting of the tables, in many cases the formatting can be improved to create a table that is easier to visually understand and incorporates more information. Note that in some cases there seems to be some general issues such as table 7 where text overlaps/
We changed the tables and tried to improve the comprehension and understanding.
- Figure 2, the peak seems to be in 2017 not 2010 as the text claims.
Text has been modified and replaced according to your suggestion. Also, we reordered Table 4 and Figure 2 to make it easier the reading. - Line 496 what is meant with “to an average of 4.2% of authors per scientific work”?
We replaced the information and make for clear the definition. There is an average of 4.2% of authors per document, which means that the 50 HCPs have been produced mostly by teams of 4 researchers.
Thank you very much. Best regards.

Round 2
Reviewer 2 Report
First off, I really appreciate the authors efforts in revising the paper. Please note that though my review was already lengthy last time, I did not provide an exhaustive list of issues as I figured it was more useful to mention the underlying general problems rather than each specific example. Perhaps because of this, the authors have mainly revised specific things mentioned but didn’t revise the paper as a whole. I should have maybe made this clearer, so for this I apologize. Nonetheless, the fact is that many underlying problems I pointed out last time (e.g. a lack of clarity behind the theoretical reasoning for the paper) remain unclear. In addition, given the very short time given to reviewers at this journal I assume that the authors have a similarly tight schedule. I think the paper can significantly improve by having the authors be able to take more time to not only revise but also rethink the paper in a more holistic way. In some ways, I think the paper may have just been submitted too fast, and I think it would be good if it develops more outside of peer review. I want to emphasize that I do see potential in the work of the authors, and given the way they dealt with my previous comments I do think the authors are able to revise the paper to a high standard. I’m mainly questioning how many rounds of revisions would be necessary to get the paper there and, as noted, think it may be beneficial for the paper to have the authors maybe take a step back and reconsider the paper as a whole, revise, and then submit as new paper. Rather than doing this through multiple round of peer reviews, where also I (or we) as reviewers may get too stuck into the details.
To help the authors with this I still list some suggestions/points below:
Major:
- though the research fields are more clearly described and introduced now in the paper, it is still not clear why all 3 need to be studied together/simultaneously as the authors do here. As this seems to hit the main purpose of the paper this remains an issue.
- More information is given now in most places, this certainly improves the paper but also often raises new questions, e.g. :
a) lines 79-85, it seems like these metrics are focused on assessing individual authors, this is somewhat confusing as the authors plan to check a research field not single authors. As they also state themselves in 86-89. It remains unclear how these metrics can be applied to a field rather than individual authors. It seems like the H-classic score is somehow a combination of the H-core and the H-index but this is also stull unclear. This is an issue as this is again a crucial part of the paper.
b) Lines 99-102, this again touches the reasoning behind the paper. I think the authors rightfully claim that this has not been done, but just because something has not been done is not enough reason that this SHOULD be done. What is still lacking in the paper is a reasoning as to why this study SHOULD be done.
c) “Computing the H-core. This phase consists of recovering the HCPs that are included in the H-core of the research area,” lines 169-170, im afraid this seems somewhat tautological, how can the H-core be computed based on HCPs that are included in the H-core? It seems that one would need to know the H-core before the HCPs that are part of this can be recovered? - Though the methods now have more details, the theoretical reasoning behind why these analyses are done remains unclear.
- In some cases, the “incorporation” of my comments is somewhat a copy/paste approach, though I think mentioning these issues does already improve the paper, I do think in many cases a real integration and consideration would make more sense. As now, just “pasted” into the text it seems often unclear what the authors really want to say. A notable exception is the part on authorship, lines 638 – 658. Here the authors really worked with the comments, included relevant references and the lines of reasoning is in principle clear. This makes me confident that the authors can also make revisions for similar points to a similar high standard.
Minor stuff:
- HCP is already used several times before it’s definition comes. This is confusing. I think this is a result of the rewriting but it would need to be rectified in a next version.
- First it’s 919 then it’s 907 punlications gathered? I guess you found 919 and then 907 were considered applicable? It would be good to give the information why the 12 studies were considered not applicable.
Author Response
First off, I really appreciate the authors efforts in revising the paper. Please note that though my review was already lengthy last time, I did not provide an exhaustive list of issues as I figured it was more useful to mention the underlying general problems rather than each specific example. Perhaps because of this, the authors have mainly revised specific things mentioned but didn’t revise the paper as a whole. I should have maybe made this clearer, so for this I apologize. Nonetheless, the fact is that many underlying problems I pointed out last time (e.g. a lack of clarity behind the theoretical reasoning for the paper) remain unclear. In addition, given the very short time given to reviewers at this journal I assume that the authors have a similarly tight schedule. I think the paper can significantly improve by having the authors be able to take more time to not only revise but also rethink the paper in a more holistic way. In some ways, I think the paper may have just been submitted too fast, and I think it would be good if it develops more outside of peer review. I want to emphasize that I do see potential in the work of the authors, and given the way they dealt with my previous comments I do think the authors are able to revise the paper to a high standard. I’m mainly questioning how many rounds of revisions would be necessary to get the paper there and, as noted, think it may be beneficial for the paper to have the authors maybe take a step back and reconsider the paper as a whole, revise, and then submit as new paper. Rather than doing this through multiple round of peer reviews, where also I (or we) as reviewers may get too stuck into the details.
To help the authors with this I still list some suggestions/points below:
Major:
- though the research fields are more clearly described and introduced now in the paper, it is still not clear why all 3 need to be studied together/simultaneously as the authors do here. As this seems to hit the main purpose of the paper this remains an issue.
Thank you for pointing this out. As indicated in the paper, consumer neuroscience, neuromarketing and neuroaesthetics are key in the design process; and it was expected to find multi-disciplinary studies that involved these fields. However, we consider necessary to conduct more profound studies that consider each of these knowledge fields separately in future studies.
- More information is given now in most places, this certainly improves the paper but also often raises new questions, e.g. :
a) lines 79-85, it seems like these metrics are focused on assessing individual authors, this is somewhat confusing as the authors plan to check a research field not single authors. As they also state themselves in 86-89. It remains unclear how these metrics can be applied to a field rather than individual authors. It seems like the H-classic score is somehow a combination of the H-core and the H-index but this is also stull unclear. This is an issue as this is again a crucial part of the paper.
This is correct, the definition of H-Classic was rephrased for clarification purposes. H-Classics aims to summarize the scientific production in a knowledge domain; but it is based on the H-Index and H-Core that quantifies individual’s research output.
b) Lines 99-102, this again touches the reasoning behind the paper. I think the authors rightfully claim that this has not been done, but just because something has not been done is not enough reason that this SHOULD be done. What is still lacking in the paper is a reasoning as to why this study SHOULD be done.
The introduction provides several reasons that support the necessity to conduct this study. Firstly, there is an ongoing controversy regarding the relationship between consumer neuroscience, neuromarketing and neuroesthetics, with numerous publications in this field endorsing various points of view. Therefore, a way to synthesize the most relevant research in this field is needed. Moreover, the study of classical articles provides a clear view of the past, present and possible future of a specific knowledge area.
c) “Computing the H-core. This phase consists of recovering the HCPs that are included in the H-core of the research area,” lines 169-170, im afraid this seems somewhat tautological, how can the H-core be computed based on HCPs that are included in the H-core? It seems that one would need to know the H-core before the HCPs that are part of this can be recovered?
Thank you for pointing this out. We have rephrased the explanation of how the H-Core was obtained.
- Though the methods now have more details, the theoretical reasoning behind why these analyses are done remains unclear.The scientometric analysis performed on this research followed a well-known H-Classics analysis approach that has been applied to previous studies, references 25 – 31.
- In some cases, the “incorporation” of my comments is somewhat a copy/paste approach, though I think mentioning these issues does already improve the paper, I do think in many cases a real integration and consideration would make more sense. As now, just “pasted” into the text it seems often unclear what the authors really want to say. A notable exception is the part on authorship, lines 638 – 658. Here the authors really worked with the comments, included relevant references and the lines of reasoning is in principle clear. This makes me confident that the authors can also make revisions for similar points to a similar high standard.
Thank you for pointing this out. In some cases, we considered the reviewer’s opinion to accurately fit the authors point of view, so we included some of these suggestions in the manuscript.
Minor stuff:
- HCP is already used several times before it’s definition comes. This is confusing. I think this is a result of the rewriting but it would need to be rectified in a next version. We have included a brief explanation about the HCPs in the introduction section.
- First it’s 919 then it’s 907 punlications gathered? I guess you found 919 and then 907 were considered applicable? It would be good to give the information why the 12 studies were considered not applicable.Thank you for pointing this out. It was a typo, there were 907 publications gathered. This is now corrected in the manuscript.
